# Institutional Trust as a Protective Factor during the COVID-19 Pandemic in China

**DOI:** 10.3390/bs12080252

**Published:** 2022-07-25

**Authors:** Shuangshuang Li, Yijia Sun, Jiaqi Jing, Enna Wang

**Affiliations:** 1Faculty of Education, Henan Normal University, Xinxiang 453007, China; sunyijia0324@163.com (Y.S.); jjq200108@163.com (J.J.); 2School of Education, Tianjin University, Tianjin 300072, China

**Keywords:** institutional trust, belief in a just world, fear of COVID-19, subjective well-being

## Abstract

Previous research has demonstrated that institutional trust protects subjective well-being during pandemics. However, the potential mediation mechanisms underlying this linkage remain unclear. This study constructs a mediating model to investigate the effect of institutional trust on subjective well-being and the mediating roles of belief in a just world and fear of COVID-19 in the Chinese context. To this end, we survey a sample of 881 participants. The results show that institutional trust, belief in a just world, fear of COVID-19, and subjective well-being (i.e., life satisfaction, positive affect, and negative affect) are significantly interrelated. The results also indicate a significant impact of institutional trust on life satisfaction, positive affect, and negative affect. Belief in a just world and fear of COVID-19, independently and in sequence, mediate the relationship between institutional trust and subjective well-being.

## 1. Introduction

The coronavirus disease that emerged in 2019 (COVID-19) has caused a health pandemic and still is one of the world’s most prominent issues today. The World Health Organization has reported, since the outbreak, over 523 million confirmed cases of COVID-19 and more than 6.2 million deaths [1]. The COVID-19 outbreak has severely threatened people’s physical and mental health [2,3,4,5]. Individuals have faced the worry of being infected and the pressure of interpersonal isolation during the COVID-19 crisis, resulting in increased depression and fear [6], severe sleep disruptions [7], and decreased subjective well-being [8]. Therefore, how individuals cope with the ongoing fear of COVID-19 and sustain their subjective well-being has become a crucial concern [9].

Institutional trust is pivotal in protecting subjective well-being during pandemics and disasters [6,10,11]. However, the mechanisms through which institutional trust relates to or influences subjective well-being in the context of the COVID-19 crisis remain unclear. This study addresses this research gap by focusing on the mediating roles of belief in a just world and fear of COVID-19 in the relationship between institutional trust and subjective well-being among the Chinese population.

Subjective well-being, a primary mental health index, is usually described as cognitive and emotional evaluations of different aspects of life and involves life satisfaction and affects [12]. Prior research has demonstrated that institutional trust substantially contributes to subjective well-being [13,14,15]. The logic behind this link relates to theoretical accounts of social capital [16,17]. Institutional trust, a cognitive component of social capital, reflects the security and confidence of the individual in particular institutions such as the government, media, health institutions, and neighborhoods [15]. On the one hand, a high level of institutional trust mirrors sufficient social support and available resources, which prevent individuals from feeling powerless and helpless within a community and foster their subjective well-being [18]. On the other hand, institutional trust may improve quality of life and mental wellness by encouraging social participation and decreasing depression, anxiety, and stress [19]. Empirical research confirms these theoretical assumptions. For example, Ciziceno and Travaglino show that perceived trust in institutions such as the healthcare system and the government is significantly associated with life satisfaction [13]. Lee reveals that older adults’ institutional trust plays an essential role in buffering against the negative well-being outcomes of the COVID-19 [11]. Matsushima et al. have surveyed postpartum women during the COVID-19 pandemic, indicating that higher trust in neighbors is related to lower negative affect, namely, depression and fear [6]. Recent research shows that trust in government is significantly associated with Chinese participants’ depressive symptoms and life satisfaction [20]. These studies suggest that institutional trust positively impacts individual subjective well-being, affecting life satisfaction and affect during the COVID-19 pandemic.

Institutional trust likely depends on belief in a just world, affecting subjective well-being indirectly. Belief in a just world has been defined as individuals’ views that they live in a just, stable, orderly, and predictable world, in which people receive what they deserve [21]. On the one hand, Zhang and Zhang propose that public institutions’ perceived trustworthiness helps Chinese people believe that the physical and social environments are just [15]. Empirical research has shown that trust in institutions (such as the government, employing units, and communities) is significantly associated with a firm belief in a just world [22,23]. On the other hand, the theoretical literature highlights that the belief in a just world prevents an individual from experiencing negative feelings, such as anxiety and depression, allowing people to feel negative life events less personally [24]. Those with a solid belief in a just world also have a lower perception of risk and interpret events from a positive perspective, maintaining subjective well-being even when they face hardships or stressful events. They tend to believe that they do not deserve negative outcomes [25]. Many empirical studies support a positive relationship between belief in a just world and subjective well-being [26,27,28]. For instance, the research by Li and Li shows that high level of belief in a just world is related to Chinese adolescents’ subjective well-being [29]. Li and colleagues also report that belief in a just world makes significant contributions to Chinese university students’ life satisfaction and positive affect [30]. A recent study has found that belief in a just world may protect healthy Chinese citizens’ positive affect and reduce negative affect during the COVID-19 crisis [31]. Therefore, it seems reasonable to assume that institutional belief may influence individuals’ subjective well-being, most likely via belief in a just world in the Chinese context.

An additional and unique perspective links institutional trust and subjective well-being through the fear of COVID-19, caused by infectious spread, unexpected lockdown, and social distancing [32]. According to Legido-Quigley et al. and Yuan et al., trust in the government and health institutions may weaken the COVID-19 fear by increasing participants’ willingness to comply with the official instructions or commands, such as using face masks and testing for COVID-19 [33,34]. Matsushima et al. show that those trusting that their neighbors follow infection control measures exhibit less fear of COVID-19 [6]. The COVID-19 fear may induce a generalized negative affect [35,36]. For example, Chen and colleagues have collected data from 2445 Chinese students and found that fear of COVID-19 is associated with generalized feelings of strain, depression, and anxiety [37]. This finding is further replicated by research based on a sample of Chinese primary school students, middle school students, and adults [38]. Recent studies demonstrate that the fear of COVID-19 negatively predicts life satisfaction or well-being [3,39]. Based on the above evidence, we infer that the fear of COVID-19 bridges the relationship between institutional trust and subjective well-being. Solid institutional trust may reduce the fear of COVID-19, thus maintaining well-being in the Chinese context.

Furthermore, we contend that belief in a just world and the fear of COVID-19 (in sequence) may mediate the relationship between institutional trust and subjective well-being. Individuals with a firm belief in a just world assume that they may protect against COVID-19 if they adhere to preventive measures [40]. Hence, belief in a just world may help overcome the fear of COVID-19. Furthermore, previous research based on Chinese samples indicates that belief in a just world reduces the risk of fear and anxiety during a disaster such as an earthquake [41,42]. Moreover, an experimental study conducted in China has shown that belief in a just world reduces the fear and anxiety of COVID-19 [31]. Therefore, we contend that institutional trust indirectly affects subjective well-being through the serial mediation of belief in a just world and fear of COVID-19.

Correctly identifying predictors of individual well-being and their underlying mechanisms of action is essential to target intervention against COVID-19. Although the critical role of institutional trust for subjective well-being has already been documented in the context of the COVID-19 pandemic, its mechanism of action has not been clearly identified.

The purposes of the current study are twofold: (a) to explore the protective effect of institutional trust on Chinese people’s subjective well-being during the COVID-19 pandemic, and (b) to test whether belief in a just world and fear of COVID-19 mediate the relation between institutional trust and subjective well-being among the Chinese population. To this end, we collect data on participants’ institutional trust, belief in a just world, fear of COVID-19, and subjective well-being (i.e., life satisfaction, positive affect, and negative affect) from China. Then, we construct mediating model to investigate the effects and mediating mechanisms of institutional trust. The study’s findings provide meaningful insights into the indirect effects of institutional trust on subjective well-being and support its role in alleviating the negative consequences of the COVID-19 pandemic. Based on a synthesis of relevant theoretical perspectives and empirical research, we developed the following research hypotheses to guide our data analysis and interpretation of results.

**Hypothesis** **1** **(H1).**
*Institutional trust is significantly associated with subjective well-being.*


**Hypothesis** **2** **(H2).**
*Belief in a just world mediates the association between institutional trust and subjective well-being.*


**Hypothesis** **3** **(H3).**
*Fear of COVID-19 mediates the association between institutional trust and subjective well-being.*


**Hypothesis** **4** **(H4).**
*Belief in a just world and fear of COVID-19 play a sequential mediatorship role in the relation between institutional trust and subjective well-being.*


## 2. Materials and Methods

### 2.1. Study Design and Data Collection

To estimate the effect of institutional trust on subjective well-being and the mediating effect of belief in a just world and fear of COVID-19. The online-based questionnaire survey approach was adopted in this study for data collection. This cross-sectional investigation was conducted in Zhengzhou city from 2–21 August 2021, when the city applied strict social distancing rules. Thus, an online data collection tool is more appropriate during the COVID-19 lockdown [3,9,37]. All the constructs in the current research were measured by structured scales that have been repeatedly used in different Chinese populations with good psychometric properties [15,37,42,43,44]. First, we delivered the study questionnaire electronically via a professional online survey platform. Convenience sampling and snowball sampling techniques were used to collect data in a cost-effective manner [7,11,37]. That is, we asked the initial participants to pass on the questionnaire through their social networks. Second, participants were informed clearly that the survey was anonymous and voluntary and their personal information was kept in confidence. Then, 881 participants (351 men and 530 women; *M_age_* = 35.15, *SD* = 5.55) completed the online survey. More than half of the participants held academic degrees (*n* = 537, 60.95%). Electronic informed consent was obtained from the participants before their participation in the formal survey. The Institutional Review Board of the corresponding author’s university approved this research.

### 2.2. Measurement

We measured trust in public institutions (i.e., central government, local government, social security institutions, health institutions, mass media, employing units, communities, and neighborhoods) using the Chinese version of the *Institutional Trust Scale* [15]. This questionnaire includes eight items rated on a five-point scale (from one = very distrustful to five = very trustful). Higher scores indicate a higher level of institutional trust. In the current study, the internal consistency is 0.86.

We assessed belief in a just world using the Chinese version of the *General Belief in a Just World Scale* [42]. Participants rated six items on a six-point Likert scale (from one = strongly disagree to six = strongly agree), with higher scores indicating a higher level of belief in a just world. In the present study, the internal consistency is 0.88.

We used the seven-item *Fear of COVID-19 Scale* (FCV-19S) to assess the fear of COVID-19 [2]. We recorded responses to these items using a five-point Likert-type scale, ranging from one (strongly disagree) to five (strongly agree). Higher scores indicate a greater fear of COVID-19. Previous research has reported that the Chinese version is acceptable and reliable [37,38]. In the current study, the internal consistency of FCV-19S equals 0.88.

We measured subjective well-being using *the Satisfaction with Life Scale* (SWLS) and *the Positive Affect and Negative Affect Scale* (PANAS) [45,46]. The SWLS includes five items rated on a seven-point Likert scale (one = strongly disagree, seven = strongly agree). The PANAS comprises a 10-item positive affect scale and a 10-item negative affect scale, rated on a five-point Likert scale (one = never, five = always). These Chinese-version scales are acceptable and reliable [43,44]. In this study, the internal consistency of SWLS, positive affect scale, and negative affect scale are equal to 0.88, 0.89, and 0.93, respectively.

### 2.3. Data Analysis

The research hypotheses were tested in three steps. First, Cronbach’s alpha and confirmatory factor analysis were employed to examine reliability and validity of all scales for measuring the constructs, respectively. Second, we calculated the descriptive statistics (i.e., mean, standard deviation, skewness, and kurtosis) and bivariate Pearson correlations. Third, the mediating effects of belief in a just world and fear of COVID-19 on the relationship between the institutional trust and subjective well-being were examined using 5000 bootstrap samples with confidence intervals of 95%. In the analysis, demographic variables, i.e., gender, age, and education, were included as covariates.

## 3. Results

### 3.1. Preliminary Analysis and Descriptive Statistics

We performed confirmatory factor analysis to assess all measures’ construct and discriminant validity. The measurement model comprised the following latent variables: institutional trust, belief in a just world, fear of COVID-19, life satisfaction, positive affect, and negative affect. We treated the items of each measure as manifest variables. The six factors model guarantees a good degree of fit, and all items have significant loadings on their latent factor: *χ*^2^/*df* = 3.696 < 5, CFI = 0.908 > 0.90, TLI = 0.902 > 0.90, RMSEA = 0.055 < 0.08, and SRMR = 0.053 < 0.08.

Table 1 presents the means, standard deviations, skewness, kurtosis, and bivariate correlations for all measures. We find significant correlations between institutional trust and subjective well-being measurements (life satisfaction: *r* = 0.240, positive affect: *r* = 0.344, negative affect: *r* = −0.255), indicating that the higher level of institutional trust is related to higher level of subjective well-being. Belief in a just world is significantly related to institutional trust (*r* = 0.332) and subject well-being measurements (life satisfaction: *r* = 0.237, positive affect: *r* = 0.230, negative affect: *r* = −0.223), suggesting that higher level of belief in a just world is associated with a higher level of institutional trust and subjective well-being. The results also indicated that the fear of COVID-19 is significantly correlated with institutional trust (*r* = −0.258), belief in a just world (*r* = −0.164), and subjective well-being measurements (life satisfaction: *r* = −0.270, positive affect: *r* = −0.216, negative affect: *r* = 0.443). It demonstrates that increased fear of COVID-19 is associated with decreased institutional trust, belief in a just world, and subjective well-being.

### 3.2. Serial Multiple Mediational Analyses

To examine the sequential mediation of belief in a just world and fear of COVID-19 in the relationship between institutional trust and life satisfaction, positive affect, and negative affect, respectively, we constructed three mediational models (see Figure 1). In addition, we specified gender, age, and education as covariate variables. We applied the bias correction bootstrap technique based on 5000 samples to test the total, direct, and indirect effect. As Hayes indicates, the effect is significant if the 95% confidence intervals do not comprise zero [47].

Institutional trust directly predicts life satisfaction (total effect, *β* = 0.244, 95% CI = [0.176, 0.309], positive affect (total effect, *β* = 0.349, 95% CI = [0.277, 0.422]), and negative affect (total effect, *β* = −0.252, 95% CI = [−0.326, −0.172]) at a moderate level. When we consider the mediation variables, as shown in Figure 1, institutional trust still has significant impact on life satisfaction (direct effect, *β* = 0.137, 95% CI = [0.055, 0.220]), positive affect (direct effect, *β* = 0.276, 95% CI = [0.187, 0.365]), and negative affect (direct effect, *β* = −0.113, 95% CI = [−0.185, −0.048]). It indicates that institutional trust elicits moderate direct effects on subjective well-being measures. In addition, institutional trust is a direct and significant predictor of belief in a just world (*β* = 0.327, 95% CI = [0.254, 0.398]) and fear of COVID-19 (*β* = −0.220, 95% CI = [−0.296, −0.134]), suggesting that institutional trust predicts belief in a just world and fear of COVID-19 at moderate levels.

Table 2 illustrates the results of the mediation test. First, institutional trust is an indirect predictor of life satisfaction (*β* = 0.046, 95% CI = [0.022, 0.080]), positive affect (*β* = 0.037, 95% CI = [0.012, 0.066]), and negative affect (*β* = −0.039, 95% CI = [−0.065, −0.014]) through belief in a just world. It suggests that institutional trust predicts subjective well-being indirectly through belief in a just world at a small level. Likewise, fear of COVID-19 mediates the effect of institutional trust on life satisfaction (*β* = 0.051, 95% CI = [0.025, 0.081]), positive affect (*β* = 0.031, 95% CI = [0.011, 0.057]), and negative affect (*β* = −0.087, 95% CI = [−0.121, −0.055]). Such results demonstrate that institutional trust predicts subjective well-being indirectly via fear of COVID-19 at a small level. In addition, institutional trust affects each subjective well-being measure via belief in a just world and fear of COVID-19 in a sequential manner at a small level (life satisfaction, *β* = 0.007, 95% CI = [0.001, 0.014]; positive affect, *β* = 0.005, 95% CI = [0.001, 0.010]; negative affect, *β* = −0.013, 95% CI = [−0.023, −0.003]).

## 4. Discussion

The goal of the current study was to investigate the positive effect of institutional trust on subjective well-being during the COVID-19 crisis. Furthermore, we particularly focused on examining whether belief in a just world and fear of COVID-19 mediated the relationship between institutional trust and subjective well-being. This study provides novel contribution to the literature as empirical research has not yet explored the mediation mechanism undying the beneficial impact of institutional trust on mental health during the COVID-19 pandemic. To this end, we collect new data on self-reported institutional trust, belief in a just world, fear of COVID-19, and subjective well-being from a sample of 881 Chinese citizens. Our primary findings confirm that institutional trust contributes significantly to life satisfaction, positive affect, and negative affect but they also reveal that belief in a just world and fear of COVID-19 play a sequential mediational role in these linkages.

The study’s results show significant correlations between institutional trust and subjective well-being (i.e., life satisfaction, positive affect, and negative affect), indicating that those who trust in public institutions exhibit higher levels of life satisfaction, positive affect, and lower levels of negative affect. This finding is consistent with theoretical perspectives highlighting the crucial role of institutional trust in subjective well-being [17]. According to the theoretical social capital model, institutional trust is a critical social resource for subjective well-being as it helps individuals maintain stable social relationships and gain support from public institutions [16,17]. Moreover, when an individual faces adverse environmental challenges, such as disasters or pandemics, institutional trust may foster people’s security feelings and help-seeking behaviors as they believe that public institutions are working for the best interests of the population and operating effectively. These beliefs are crucial for individual subjective well-being [11,48]. Our results also align with existing experimental studies suggesting that positive attitudes toward institutional relationships are significantly associated with better life satisfaction, more positive affect, and less negative affect during the COVID-19 crisis [6,11,49].

The mediation effect analysis indicates that, in line with our expectations, high institutional trust increases individual subjective well-being through belief in a just world. This result echoes the assumptions of Zhang and Zhang that the belief in a just world depends on a safe and trusted environment, and mistrust in social institutions contributes to the belief that the world is unreliable, uncontrollable, and unpredictable, decreasing the quality of life and subjective well-being [15]. Our findings are consistent with previous studies indicating a close link between institutional trust and individual belief in a just world. This result is in line with prior research demonstrating that belief in a just world has a protective effect on subjective well-being [26,27,28]. Furthermore, higher institutional trust reflects higher satisfaction with institutions, which further influences individual confidence and attitude toward society [15]. Individuals with a positive attitude and strong confidence in social institutions tend to believe that the world is just, orderly, reliable, organized, and safe [50]. This generalized positive view may increase the experience of positive affect and life satisfaction and reduce the likelihood of negative affect [27,28].

Institutional trust also affects subjective well-being through the fear of COVID-19. Previous research has suggested that solid institutional trust is critical in alleviating the fear of COVID-19 [6]. Individuals with high institutional trust are more likely to believe that public institutions may take adequate measures to curb the spread of highly infectious diseases, such as COVID-19. A recent study based on Chinese sample shows that a higher level of institutional trust led more people follow infection control measures such as proper preventive and help seeking behaviors [51]. Hence, adhering to infection control instructions may decrease their fear of COVID-19 [34]. Individuals with higher fear of COVID-19 may experience more general negative affect and perceive a lower level of life satisfaction or well-being [3,39]. Therefore, a solid institutional trust may protect subjective well-being by decreasing the fear of COVID-19.

In addition, we show that the serial effect of belief in a just world and fear of COVID-19 mediate the relationship between institutional trust and subjective well-being. This result contributes to a deeper understanding of how institutional trust has protected individual mental health during the COVID-19 crisis. Confidence in social institutions may contribute to the general view that the world is fair to everyone and people receive what they deserve (i.e., belief in a just world) [15]. Individuals with a strong belief in a just world are more likely to think those who contract COVID-19 “deserve it” as they do not adhere to epidemic prevention norms such as wearing masks and keeping social distance [40]. Thus, individuals with a firm belief in a just world are more inclined to consider the COVID-19 epidemic controllable and consciously abide by public health instructions to prevent COVID-19 infection and protect their health, reducing anxiety and the fear of COVID-19 and preserving their subjective well-being during the pandemic [31,39].

Overall, the study’s results indicate that institutional trust influences subjective well-being directly and indirectly through belief in a just world and fear of the COVID-19. To the best of our knowledge, this is one of the few studies to examine the relationships between institutional trust, subjective well-being, belief in a just world, and fear of COVID-19. From a theoretical perspective, our study offers a valid framework to explain how institutional trust affects individual subjective well-being in the context of the COVID-19 crisis. Those who trust social institutions strongly believe in a just world, which helps overcome COVID-19 fear and further increases their subjective well-being. These results enrich the current knowledge of the mediating mechanism underlying institutional trust’s buffer and protective effect for those who suffer from disasters. From a practical perspective, our study may help institutions, such as government and health institutions, realize the vital role of institutional trust during crises, such as the COVID-19 pandemic. These measures may successfully alleviate people’s fear and anxiety of COVID-19 and increase their subjective well-being.

## 5. Conclusions

During the pandemic, individuals in China may experience fear and decreased subject well-being. Our study indicates that institutional trust may assist in decreasing these negative outcomes. This paper further confirms that institutional trust protects Chinese people’s subject well-being indirectly through enhancing belief in a just world and reducing fear of COVID-19.

Along with our finding, this study has several limitations, suggesting future research directions. First, restricted to many factors such as environment, time, and sampling condition, we use convenience sampling and snowball sampling to collect data and investigate adult residents in Zhengzhou city through anonymous online survey. Such procedures do not allow us to consider our sample as a representative one, even though we statistically controlled for the demographic variables (i.e., gender, age, and education) and thus our findings may apply to the general Chinese adults. Future research could further employ other sampling methods such as random sampling and collect more nationally representative data, which would increase the robustness of our study results. Second, we mainly focus on the protective role of institutional trust in the Chinese context. However, there is also the possibility of cultural differences in the measure of institutional trust and its influence on individuals’ response to COVID-19 [11]. That is, cultural factors should be considered in future study. For example, future research could carry out a worldwide survey to explore and compare the mediating mechanisms underlying the relationship between institutional trust and subject well-being across different countries. Third, due to the study’s cross-sectional nature, we do not disentangle the causal relationships between the four variables of interest. Nevertheless, even though we should be cautious about the causal interpretation of our findings, previous literature sustains the mediating effect we investigated. Future longitudinal or experimental studies should explore the influencing factors and mechanisms of action of subjective well-being during the COVID-19 pandemic. Finally, the present analysis mainly employs self-reported measures, which may produce common-method bias and affect intercorrelations between the measures. The confirmatory factor analysis indicates that all measures present a good discriminant validity, suggesting that common-method bias is not a significant problem in our study [52]. Future research may combine other methods such as observations and interviews to control common-method bias and test the proposed mediating model.

## Figures and Tables

**Figure 1 behavsci-12-00252-f001:**
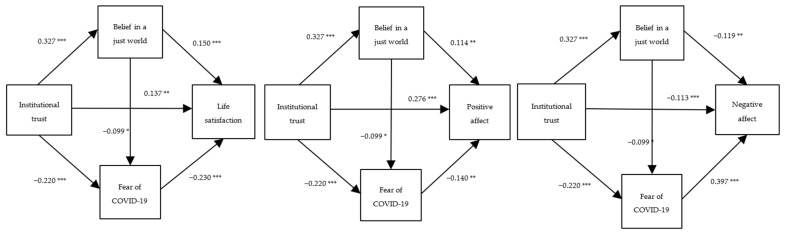
The results of serial multiple mediational models, * *p* < 0.05, ** *p* < 0.01, and *** *p* < 0.001. Values shown are standardized coefficients.

**Table 1 behavsci-12-00252-t001:** Means, standard deviations, skewness, kurtosis, and bivariate correlations among measures.

Measure	*M*	*SD*	Skewness	Kurtosis	1	2	3	4	5
1. Institutional trust	34.96	5.33	−1.38	2.88					
2. Belief in a just world	25.48	5.50	−0.30	0.84	0.33 ***				
3. Fear of COVID-19	20.03	7.30	0.21	−0.62	−0.26 ***	−0.16 ***			
4. Life satisfaction	21.78	6.15	−0.02	0.07	0.24 ***	0.24 ***	−0.27 ***		
5. Positive affect	32.46	6.58	−0.27	1.37	0.34 ***	0.23 ***	−0.22 ***	0.47 ***	
6. Negative affect	27.50	8.62	−0.01	−0.23	−0.26 ***	−0.22 ***	0.44 ***	−0.09 **	0.011

** *p* < 0.01, *** *p* < 0.001.

**Table 2 behavsci-12-00252-t002:** Indirect effect of institutional trust on subjective well-being via belief in a just world and fear of COVID-19.

Path	Standardized Coefficient	95% CI	
		LL	UL
** *Model 1* **			
Institutional trust → BJW → Life satisfaction	0.049	0.022	0.080
Institutional trust→ Fear of COVID-19 → Life satisfaction	0.051	0.025	0.081
Institutional trust → BJW → Fear of COVID-19 → Life satisfaction	0.007	0.001	0.014
Total effect	0.244	0.176	0.309
Direct effect	0.137	0.055	0.220
Total indirect effect	0.107	0.067	0.150
** *Model 2* **			
Institutional trust → BJW → Positive affect	0.037	0.012	0.066
Institutional trust → Fear of COVID-19 → Positive affect	0.031	0.011	0.057
Institutional trust → BJW → Fear of COVID-19→ Positive affect	0.005	0.001	0.010
Total effect	0.349	0.277	0.422
Direct effect	0.276	0.187	0.365
Total indirect effect	0.073	0.040	0.111
** *Model 3* **			
Institutional trust → BJW → Negative affect	−0.039	−0.065	−0.014
Institutional trust → Fear of COVID-19 → Negative affect	−0.087	−0.121	−0.055
Institutional trust → BJW → Fear of COVID-19 → Negative affect	−0.013	−0.023	−0.003
Total effect	−0.252	−0.326	−0.172
Direct effect	−0.113	−0.185	−0.048
Total indirect effect	−0.139	−0.180	−0.098

CI, confidence interval; LL, lower limit; UL, upper limit; BJW, belief in a just world.

## Data Availability

Data available on request from the authors.

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
