# Peer review of "Institutional Trust as a Protective Factor during the COVID-19 Pandemic in China"

_behavsci, 2022, doi:10.3390/bs12080252_

Round 1

Reviewer 1 Report

This work makes a contribution to the literature in reaffirming that institutional trust can promote subjective feelings of well-being.

The literature review is strong. The sample size is of 881 excellent. The Materials and Methods section is very clear and strong.

I hope that at least one reviewer has statistical expertise since I do not.

The title is much too long and cumbersome and needs to be shortened. The authors can consider: "Institutional Trust as a Protective Factor during the Covid-19 Pandemic in China."

With the exception of the Materials and Methods section, there is hardly any reference to the study taking place in China. This is important to include in the abstract and throughout the article.

The major concern about this article is that there seems to be an assumption that "institutional trust" means the same thing in all countries. Is this true? For example, is the reader to assume that "institutional trust" operates the same way in China, the United States, Mexico, and Brazil? There have been such disparate responses to the pandemic based on culture and politics in different countries that I think the authors must find some way to address these concerns.

I suggest the authors consider a section titled Conclusion. The current last paragraph lines 268-281 seem to fit better at the end of the Discussion section. The Conclusion can begin with line 259 "the study has several limitations....". the rest of the paragraph is good as written but is very general. What more can be said about using random sampling and other methods? I'd like to see the authors make more specific recommendations that can guide their future research as well as the research of others.

Author Response

Response to Reviewer 1 Comments

This work makes a contribution to the literature in reaffirming that institutional trust can promote subjective feelings of well-being.

The literature review is strong. The sample size is of 881 excellent. The Materials and Methods section is very clear and strong.

I hope that at least one reviewer has statistical expertise since I do not.

We are grateful to the Reviewer for taking the time to review our manuscript. We appreciate the comments and advice, which have helped us to improve the content and clarity of the manuscript. We have revised the manuscript taking into account the suggestions of the Reviewer. Please see below, in red, for a point-by-point response to the comments and concerns. All page numbers refer to the revised manuscript file with tracked changes. All modifications in the manuscript have been highlighted in red.

Point 1: The title is much too long and cumbersome and needs to be shortened. The authors can consider: "Institutional Trust as a Protective Factor during the Covid-19 Pandemic in China."

Response 1: Thank you for pointing this out. We revised the title as suggested by the reviewer. Please see our revised manuscript.

Point 2: With the exception of the Materials and Methods section, there is hardly any reference to the study taking place in China. This is important to include in the abstract and throughout the article.

Response 2: Thanks for your nice reminder. We have cited some studies based on Chinese samples in introduction and discussion sections in the original manuscript. For example, studies by Zhang et al. (2015) and Jiang et al. (2016) support the mediating role of the belief in a just world. We also cited research by Wang et al. (2021), Ye et al. (2020), Xie et al. (2011) and Wu et al. (2009) in order to support the view that institutional trust predict subjective well-being through belief in a just world and fear of COVID-19 in sequence.

In addition, we further added more studies based on Chinese sample (see lines 51-53, lines 70-75 and lines 85-89 on page 2, and lines 287-289 on page 7) to the manuscript.

  1. Zhang, Z.; Zhang, J. Belief in a just world mediates the relationship between institutional trust and life satisfaction among the elderly in China. Individ. Differ. 2015, 83, 164–169. doi:10.1016/j.paid.2015.04.015.
  2. Jiang, F.; Yue, X.; Lu, S.; Yu, G.; Zhu, F. How belief in a just world benefits mental health: The effects of optimism and gatitude. Indic. Res. 2016, 126, 411–423. doi:10.1007/s11205-015-0877-x.
  3. Wang, J.; Wang, Z.; Liu, X.; Yang, X.; Zheng, M.; Bai, X. The impacts of a COVID-19 epidemic focus and general belief in a just world on individual emotions. Individ. Differ. 2021, 168, 110349. doi:10.1016/j.paid.2020.110349.
  4. Ye, B.; Wu, D.; Im, H.; Liu, M.; Wang, X.; Yang, Q. Stressors of COVID-19 and stress consequences: The mediating role of rumination and the moderating role of psychological support. Youth Serv. Rev. 2020, 118, 105466. doi:10.1016/j.childyouth.2020.105466.
  5. Xie, X.; Liu, H.; Gan, Y. Belief in a just world when encountering the 5/12 Wenchuan earthquake. Behav. 2011, 43, 566–586. doi:10.1177/0013916510363535.
  6. Wu, S.T.; Wang, L.; Zhou, M.; Wang, W.; Zhang, J. Belief in a just world and subjective well-being: Comparing disaster sites with normal areas. Psychol. Sci. 2009, 17, 579–587. doi:CNKI:SUN:XLXD.0.2009-03-024.

Point 3: The major concern about this article is that there seems to be an assumption that "institutional trust" means the same thing in all countries. Is this true? For example, is the reader to assume that "institutional trust" operates the same way in China, the United States, Mexico, and Brazil? There have been such disparate responses to the pandemic based on culture and politics in different countries that I think the authors must find some way to address these concerns.

Response 3: We thank the reviewer for bringing the cultural differences into our attention. We indeed ignored this issue in the original manuscript. Previous studies demonstrated the positive effect of institutional trust on subjective well-being in many countries, such as Japan (Matsushima et al., 2021), Italy (Paolini et al., 2020), Sweden (Esaiasson et al., 2020) and other European countries (Lee, 2022). However, there is also the possibility of cultural differences in the measure of institutional trust and its influence on individuals’ response to COVID-19 (Lee, 2022). Our study focused particularly on the protective role of institutional trust among the Chinese population and we agree that our findings are limited to be generalized into other culture context.

Therefore, following the suggestion by the reviewer, we made some revisions in the manuscript. First, we revised the title, suggesting that our study focus on Chinses population. Second, we revised descriptions in introduction section, such as “The purposes of current study are twofold: (a) to explore the protective effects of institutional trust on Chinese people’s subjective well-being during the COVID-19 pandemic, and (b) to test whether belief in a just world and fear of COVID-19 mediate the relation between institutional trust and subjective well-being among the Chinese population. To this end, we collect data on participants’ institutional trust, belief in a just world, fear of COVID-19, and subjective well-being (i.e., life satisfaction, positive affect, and negative affect) from China.” (Page 3 Line 108-112). Third, we further elaborated in the conclusion section that our findings are only based on Chinses samples and ask the readers to be cautious about the interpretation until the finding is replicated in other culture context. Please see Page 8 Line 334-340: “Second, we mainly focus on the protective role of institutional trust in the Chinese context. However, there is also the possibility of cultural differences in the measure of institutional trust and its influence on individuals’ response to COVID-19 [11]. That is, cultural factors should be considered in future study. For example, future research could carry out a worldwide survey to explore and compare the mediating mechanisms underlying the relationship between institutional trust and subject well-being across different countries.”

  1. Matsushima, M.; Tsuno, K.; Okawa, S.; Hori, A.; Tabuchi, T. Trust and well-being of postpartum women during the COVID-19 crisis: Depression and fear of COVID-19. SSM - Popul. Health. 2021, 15, 100903. doi:10.1016/j.ssmph.2021.100903.
  2. Paolini, D. COVID-19 Lockdown in Italy: The role of social identification and social and political trust on well-being and distress. Curr Psychol. 2020, 1-8, doi:10.1007/s12144-020-01141-0.
  3. Esaiasson, P.; Sohlberg, J.; Ghersetti, M.; Johansson, B. How the coronavirus crisis affects citizen trust in institutions and in unknown others: Evidence from ‘the Swedish experiment’. J. Poli.t Res.. 2020. doi.: 10.1111/1475-6765.12419
  4. Lee, S. Subjective well-being and mental health during the pandemic outbreak: Exploring the role of institutional trust. Aging. 2022, 44, 10–21. doi:10.1177/0164027520975145.

Point 4: I suggest the authors consider a section titled Conclusion. The current last paragraph lines 268-281 seem to fit better at the end of the Discussion section. The Conclusion can begin with line 259 "the study has several limitations....". the rest of the paragraph is good as written but is very general. What more can be said about using random sampling and other methods? I'd like to see the authors make more specific recommendations that can guide their future research as well as the research of others.

Response 4: Thank you very much for pointing this out. Following this suggestion, we added conclusion section and revised the discussion. Please see Page 8 Line 327-334: “First, restricted to many factors such as environment, time, and sampling condition, we use convenience sampling and snowball sampling to collect data and investigate adult residents in Zhengzhou city through anonymous online survey. Such procedures do not allow us to consider our sample as a representative one, even though we statistically controlled for the demographic variables (e.g., gender, age, and education) and thus our findings may apply to the general Chinese adults. Future research could further employ other sampling methods such as random sampling and collect more nationally representative data, which would increase the robustness of our study results.”

Thanks for the constructive comments and suggestions by the reviewer.

Reviewer 2 Report

The article is a very insightful contribution to better understand the role of institutional trust as a mediating effect towards Covid-19. The paper allows that one can clearly learn a lot about the situation in a specific geographical area. On this matter, methods are well explained and the list of references is up-to-date.

Although the article is clear and its main findings are summarized at the end of the study, discussing the data with prior scholarship, I miss proper research questions or hypotheses that guide the analysis. This would help to enhance the scientific soundness of the paper.

Author Response

Response to Reviewer 2 Comments

The article is a very insightful contribution to better understand the role of institutional trust as a mediating effect towards Covid-19. The paper allows that one can clearly learn a lot about the situation in a specific geographical area. On this matter, methods are well explained and the list of references is up-to-date. 

We thank the reviewer for the detailed and constructive comments. The through review helped immensely in the shaping of the manuscript. The suggestions and comments have been closely followed and revision have been made accordingly. Please see below, in red, for a point-by-point response to the reviewer’s comments and concerns. All page numbers refer to the revised manuscript file with tracked changes. For the reviewer’s convenience, all modifications in the manuscript have been highlighted in red.

Point 1: Although the article is clear and its main findings are summarized at the end of the study, discussing the data with prior scholarship, I miss proper research questions or hypotheses that guide the analysis. This would help to enhance the scientific soundness of the paper.

Response 1: Thanks for your valuable suggestions. We add the hypotheses in the revised manuscript. Please see Page 3 Line 117-126: “Based on a synthesis of relevant theoretical perspectives and empirical research, we developed the following research hypotheses to guide our data analysis and interpretation of results.

H1: Institutional trust is significantly associated with subjective well-being.

H2: Belief in a just world mediates the association between institutional trust and subjective well-being.

H3: Fear of COVID-19 mediates the association between institutional trust and subjective well-being.

H4: Belief in a just world and fear of COVID-19 play a sequential mediatorship role in the relation between institutional trust and subjective well-being.”

Thanks for the constructive comments and suggestions by the reviewer.

Reviewer 3 Report

A very nice piece of work – well done!

I have provided some feedback and I would appreciate if you could take my comments into consideration to improve your work!

In the introduction the topic of interest is well explained and the rationale behind the present study is shown. Relevant information is provided following a structured and logical pattern which allows the purpose of the review to be clearly evident to the reader. This part sets the scene for the presented analysis and provides the background understanding. Regarding the organization, themes are fairly connected in the literature review and the paper is logically ordered and explained to the reader. There are clear transitions between the sections, so they link together.

Although the introduction is well constructed the structure of the paper to be followed is not described to the reader. In addition, since the study is actually employed in China, you could possibly consider of adding that in the title? Or possibly you could state that the study will focus on this location in the last part of your introduction.

In the material and methods part, you present interesting information regarding the research design; however, this part has many important points missing. You need to include the research philosophy used and the research approach followed in this work. Moreover, the benefits of the selected method/survey need to be shown with regards to the examined topic. Have you selected a structured or semi-structured questionnaire and why? This would be nice to be explained. You have managed to present the structure of the questionnaire, but you have not explained whether a pilot study has been conducted, how many people participated to it and what are the amendments made after the pilot survey was completed. In this section I would expect also to include information regarding the methodologies employed and presented in the results section and to address the disadvantages/limitations associated to them as well at the end of the discussion.

The sample is composed of 881 respondents which can be considered as satisfactory; however, you need to show how well these numbers eg 351 men and 530 women reflect the gender characteristic of the population?

Furthermore, you need to state how did you ensure reliability and validity in your work apart from the ethical issues that were considered?

In results part, the preliminary analysis and the descriptive statistics are well presented. You could interpret what do the negative and positive correlation coefficients actually suggest ie Increase in one variable is associated to an increase/decrease to the other variable.

Mediation analysis is well presented with the necessary information being included to all the three constructed models. Possibly you may consider of explaining the magnitude of each effect based on the betas.

A very well constructed discussion has been presented where you compare and contrast the findings with relevant literature, and you justify your results. You have managed to present the limitations of your work and further research; however, with regard to the sampling technique employed you can be more specific on the further research component. Useful recommendations have been provided; however, they could be more direct.

You have managed to use a variety of relevant, appropriate, and current sources.

Author Response

Response to Reviewer 3 Comments

A very nice piece of work – well done!

I have provided some feedback and I would appreciate if you could take my comments into consideration to improve your work!

In the introduction the topic of interest is well explained and the rationale behind the present study is shown. Relevant information is provided following a structured and logical pattern which allows the purpose of the review to be clearly evident to the reader. This part sets the scene for the presented analysis and provides the background understanding. Regarding the organization, themes are fairly connected in the literature review and the paper is logically ordered and explained to the reader. There are clear transitions between the sections, so they link together.

The comments and suggestions are very helpful for improving the manuscript. We are very grateful to the reviewer who provided them since this way we had a valuable guideline for improving the quality of the manuscript. We responded to each one of the queries proposed by the reviewer. Please find below our detailed responses to the comments together with the descriptions of the changes in the manuscript. Changes according to reviewers’ comments are highlighted in red in the manuscript.

Point 1: Although the introduction is well constructed the structure of the paper to be followed is not described to the reader.

Response 1: Thank you for pointing this out. We revised the last paragraph in the introduction section which descripted the aims of our research. Please see Page 3 Line 108-126:” The purposes of current study are twofold: (a) to explore the protective effects of institutional trust on Chinese people’s subjective well-being during the COVID-19 pandemic, and (b) to test whether belief in a just world and fear of COVID-19 mediate the relation between institutional trust and subjective well-being among the Chinese population. To this end, we collect data on participants’ institutional trust, belief in a just world, fear of COVID-19, and subjective well-being (i.e., life satisfaction, positive affect, and negative affect) from China. Then, we construct mediating model to investigate the effects and mediating mechanism of institutional trust. The study’s findings provide meaningful insights into the indirect effects of institutional trust on subjective well-being and support its role in alleviating the negative consequences of the COVID-19 pandemic. Based on a synthesis of relevant theoretical perspectives and empirical research, we developed the following research hypotheses to guide our data analysis and interpretation of results.

H1: Institutional trust is significantly associated with subjective well-being.

H2: Belief in a just world mediates the association between institutional trust and subjective well-being.

H3: Fear of COVID-19 mediates the association between institutional trust and subjective well-being.

H4: Belief in a just world and fear of COVID-19 play a sequential mediatorship role in the relation between institutional trust and subjective well-being.”

Point 2: In addition, since the study is actually employed in China, you could possibly consider of adding that in the title? Or possibly you could state that the study will focus on this location in the last part of your introduction.

Response 2: Thanks for your nice reminder. We made some revisions according to this suggestion. First, We have revised the title:“Institutional Trust as a Protective Factor during the COVID-19 Pandemic in China”. Second, as shown in Response 1, we emphasized that the purpose of this study was to explore the protective effect of institutional trust on subjective well-being in the Chinese context.

Point 3: In the material and methods part, you present interesting information regarding the research design; however, this part has many important points missing. You need to include the research philosophy used and the research approach followed in this work. Moreover, the benefits of the selected method/survey need to be shown with regards to the examined topic.

Response 3: Thanks for your valuable suggestions. We added “Study Design and Data Collection” section in the material and methods part. First, the online-based questionnaire survey approach is more appropriate in this study as the government sets certain strictions (Duong, 2020; Rsizer et al., 2021; Chen et al., 2021). Second, convenience sampling and snowball sampling were used since these techniques might be cost-effective and they have been widely utilized in psychological and social studies (Salfi et al., 2021; Lee, 2022; Chen et al., 2021). Please see Page 3 Line 128-146:” To estimate the effect of institutional trust on subjective well-being and the mediating effect of belief in a just world and fear of COVID-19. The online-based questionnaire survey approach was adopted in this study for data collection. This cross-sectional investigation was conducted in Zhengzhou city in August 2–21, 2021, when the city applied strict social distancing rules. Thus, online data collection tool during the COVID-19 lockdown is more appropriate [3,9,37]. All the constructs in current research were measured by structured scales which have been repeatedly used in different Chinese populations with good psychometric properties [15,37,42,45,46]. First, we delivered the study questionnaire electronically via a professional online survey platform. Convenience sampling and snowball sampling techniques were used to collect data in a cost-effective manner [7,11,37]. That is, we asked the initial participants to pass on the questionnaire through their social networks. Second, participants were informed clearly that the survey was anonymous and voluntary personal information was kept in confidence. Then 881 participants (351 men and 530 women; Mage = 35.15, SD = 5.55) completed the online survey. More than half of the participants held academic degrees (n = 537, 60.95%). Electronic in-formed consent was obtained from the participants before their participation the formal survey. The Institutional Review Board of the corresponding author’s university approved this research.”

  1. Duong, C.D. The impact of fear and anxiety of Covid-19 on life satisfaction: Psychological distress and sleep disturbance as mediators. Personal. Individ. Differ. 2021, 178, 110869. doi:10.1016/j.paid.2021.110869.
  2. Salfi, F.; Lauriola, M.; D'Atri, A.; Amicucci, G.; Viselli, L.; Tempesta, D.; Ferrara, M. Demographic, psychological, chronobiological, and work-related predictors of sleep disturbances during the COVID-19 lockdown in Italy. Sci. Rep. 2021, 11,11416. doi: 10.1038/s41598-021-90993-y.
  3. Reizer, A.; Geffen, L.; Koslowsky, M. Life under the COVID-19 lockdown: On the relationship between intolerance of uncertainty and psychological distress. Psychol. Trauma Theory Res. Pract. Policy. 2021, 13, 432–437. doi:10.1037/tra0001012.
  4. Lee, S. Subjective well-being and mental health during the pandemic outbreak: Exploring the role of institutional trust. Res. Aging. 2022, 44, 10–21. doi:10.1177/0164027520975145.
  5. Chen, W.; Liang, Y.; Yin, X.; Zhou, X.; Gao, R. The factor structure and rasch analysis of the Fear of COVID-19 Scale (FCV-19S) among Chinese students. Front. Psychol. 2021, 12, 678979. doi:10.3389/fpsyg.2021.678979.

Point 4: Have you selected a structured or semi-structured questionnaire and why? This would be nice to be explained. You have managed to present the structure of the questionnaire, but you have not explained whether a pilot study has been conducted, how many people participated to it and what are the amendments made after the pilot survey was completed.

Response 4: Thank you for pointing this out. We selected structured scales (i.e., Institutional Trust Scale, General Belief in a Just World Scale, Fear of COVID-19 Scale, Satisfaction with Life Scale, and the Positive Affect and Negative Affect Scale) as they have been repeatedly used in different Chinese populations with good psychometric properties. That is, previous research has demonstrated that these instruments show good the validity and reliability in the Chinese context (Zhang & Zhang, 2015; Chen et al., 2021; Chi et al., 2022; Wu et al., 2009; Wang et al., 2021; Lin et al., 2022; Jia et al., 2021). Based on these studies, we did not conduct a pilot study to test the validity and reliability of scales we selected. We are very sorry about this. However, we have invited four psychological experts to evaluate our questionnaire and procedures before starting the formal survey and our results indeed indicate that these scales have good reliability and validity among the sample in current research. The internal consistencies (Cronbach’s alpha) of all scales are above .85 and the confirmatory factor analysis indicate all scales have acceptable construct and discriminant validity. These results might suggest that our data collected through these scales are reliable and valid.

  1. Zhang, Z.; Zhang, J. Belief in a just world mediates the relationship between institutional trust and life satisfaction among the elderly in China. Personal. Individ. Differ. 2015, 83, 164–169. doi:10.1016/j.paid.2015.04.015.
  2. Wang, J.; Wang, Z.; Liu, X.; Yang, X.; Zheng, M.; Bai, X. The impacts of a COVID-19 epidemic focus and general belief in a just world on individual emotions. Personal. Individ. Differ. 2021, 168, 110349. doi:10.1016/j.paid.2020.110349.
  3. Chen, W.; Liang, Y.; Yin, X.; Zhou, X.; Gao, R. The factor structure and rasch analysis of the Fear of COVID-19 Scale (FCV-19S) among Chinese students. Front. Psychol. 2021, 12, 678979. doi:10.3389/fpsyg.2021.678979.
  4. Chi, X.; Chen, S.; Chen, Y.; Chen, D.; Yu, W.; Guo, T.; Cao, Q.; Zheng, S.; Hossain, M. M.; Stubbs, B.; Yeung, A.; Zou, L. Psychometric evaluation of the Fear of COVID-19 Scale among Chinese population. Int. J. Ment. Health Addict. 2022, 16. doi: doi: 10.1007/s11469-020-00441-7.
  5. Wu, S.T.; Wang, L.; Zhou, M.; Wang, W.; Zhang, J. Belief in a just world and subjective well-being: Comparing disaster sites with normal areas. Adv. Psychol. Sci. 2009, 17, 579–587.
  6. Lin, L. Seeking pleasure or growth? The mediating role of happiness motives in the longitudinal relationship between social mobility beliefs and well-being in college students. Personal. Individ. Differ. 2022, 184, 111170. doi:10.1016/j.paid.2021.111170
  7. Jia, N.; Li, W.; Zhang, L.; Kong, F. Beneficial effects of hedonic and eudaimonic motivations on subjective well-being in adolescents: A two-wave cross-lagged analysis. J. Posit. Psychol. 2021, 1–7, doi:10.1080/17439760.2021.1913641.

Point 5: In this section I would expect also to include information regarding the methodologies employed and presented in the results section and to address the disadvantages/limitations associated to them as well at the end of the discussion.

Response 5: Thanks for your valuable comments. First, we added the “Data Analysis” section in the material and methods part, which displays the methodologies employed and presented in the results section. Please see Page 4 Line 171-179:” The research hypotheses were tested in three steps. First, Cronbach’s alpha and confirmatory factor analysis were employed to examine reliability and validity of all scales for measuring the constructs, respectively. Second, we calculated the descriptive statistics (i.e., mean, standard deviation, skewness, and kurtosis) and bivariate Pearson correlations. Third, the mediating effects of belief in a just world and fear of COVID-19 on the relationship between the institutional trust and subjective well-being were examined using 5000 bootstrap samples with confidence intervals of 95%. In the analysis, demographic variables, i.e., gender, age, and education were included as covariates.”

Second, we revised the descriptions regarding the limitation of our research. Please see Page 8 Line 326-351:” First, restricted to many factors such as environment, time, and sampling condition, we use convenience sampling and snowball sampling to collect data and investigate adult residents in Zhengzhou city through anonymous online survey. Such procedures do not allow us to consider our sample as a representative one, even though we statistically controlled for the demographic variables (i.e., gender, age, and education) and thus our findings may apply to the general Chinese adults. Future research could further employ other sampling methods such as random sampling and collect more nationally representative data, which would increase the robustness of our study results. Second, we mainly focus on the protective role of institutional trust in the Chinese context. However, there is also the possibility of cultural differences in the measure of institutional trust and its influence on individuals’ response to COVID-19 [11]. That is, cultural factors should be considered in future study. For example, future research could carry out a worldwide survey to explore and compare the mediating mechanisms underlying the relationship between institutional trust and subject well-being across different countries. Third, due to the study's cross-sectional nature, we do not disentangle the causal relationships between the four variables of interest. Nevertheless, even though we should be cautious about the causal interpretation of our findings, previous literature sustains the mediating effect we investigated. Future longitudinal or experimental studies should explore the influencing factors and mechanisms of action of subjective well-being during the COVID-19 pandemic. Finally, the present analysis mainly employs self-reported measures, which may produce common-method bias and affect intercorrelations between the measures. The confirmatory factor analysis indicates that all measures present a good discriminant validity, suggesting that common-method bias is not a significant problem in our study [52]. Future research may combine other methods such as observations and interviews to control common-method bias and test the proposed mediating model.”.

Point 6: The sample is composed of 881 respondents which can be considered as satisfactory; however, you need to show how well these numbers eg 351 men and 530 women reflect the gender characteristic of the population?.

Response 6: Thanks for your kind reminders. As presented in Response 3, we employed convenience sampling and snowball sampling since these techniques might be cost-effective and they have been widely utilized in psychological and social studies. Thus, our sample might have a lower representation of the Chinese population. To address this issue, the gender, age, and education was used as covariates to control the influence of demographic variables. In addition, we agree this might be one of limitations and we elaborated it on Page 8 Line 329-334:” Such procedures do not allow us to consider our sample as a representative one, even though we statistically controlled for the demographic variables (i.e., gender, age, and education) and thus our findings may apply to the general Chinese adults. Future research could further employ other sampling methods such as random sampling and collect more nationally representative data, which would increase the robustness of our study results.”

Point 7: Furthermore, you need to state how did you ensure reliability and validity in your work apart from the ethical issues that were considered?

Response 7: Thank you very much for pointing this out. We added the Study Design and Data Collection section and Data Analysis section. First, the scales we selected have good reliability and validity (Page 3 Line 134-136). Second, we informed participants clearly that the survey was anonymous and voluntary and their personal information was kept in confidence (Page 3 Line 140-141), which would reduce the social desirability bias. Third, we employed Cronbach’s alpha and confirmatory factor analysis to examine reliability and validity of all scales (Page 4 Line 172-174). Then, the direct and indirect effects of institutional trust were examined by strict methods, 5000 bootstrap samples with confidence intervals of 95%, ensuring that the statistics evaluated were accurate and unbiased as much as possible (Page 4 Line 176-178). In addition, demographic variables including gender, age, and education were controlled in analysis (Page 4 Line 178-179). The above measures would ensure reliability and validity of our results.

Point 8: In results part, the preliminary analysis and the descriptive statistics are well presented. You could interpret what do the negative and positive correlation coefficients actually suggest ie Increase in one variable is associated to an increase/decrease to the other variable.

Response 8: Thanks for your kind reminders. We revised the descriptions regarding the correlation coefficients. Please see Page 5 Line 193-202, for example, ” We find significant correlations between institutional trust and subjective well-being measurements (life satisfaction: r =.240, positive affect: r =.344, negative affect: r =-.255), indicating that the higher level of institutional trust is related to higher level of subjective well-being.”

Point 9: Mediation analysis is well presented with the necessary information being included to all the three constructed models. Possibly you may consider of explaining the magnitude of each effect based on the betas.

Response 9: We thank the reviewer for pointing this out. We explained the effects of mediating models. Please see Page 5 Line 217-239, for example, ”It indicates that institutional trust elicits moderate direct effects on subjective well-being measures.”

Point 10: A very well constructed discussion has been presented where you compare and contrast the findings with relevant literature, and you justify your results. You have managed to present the limitations of your work and further research; however, with regard to the sampling technique employed you can be more specific on the further research component. Useful recommendations have been provided; however, they could be more direct.

Response 10: Thanks for your valuable suggestions. We revised the descriptions regarding the limitation of current study. As presented in Response 5, we elaborated the limitations of methods and provided more direct recommendations for future research. Please see Page 8 Line 326-351.

You have managed to use a variety of relevant, appropriate, and current sources.

Thanks for the constructive comments and suggestions by the reviewer.

Round 2

Reviewer 1 Report

The authors have done a very good job of responding to reviewer recommendations. I appreciate the additional explanation of the statistical analyses. By making clear they are focusing specifically on China, the authors can entice other researchers to examine similar issues in their own countries.

Author Response

Thanks for the constructive comments and suggestions by the reviewer.

Reviewer 3 Report

Well done to the author(s). You have managed to account all my suggestions and thank you very much for that!

I believe that the quality of your work has been improved and hope that the comments were constructive!

Author Response

(The authors gave the same response as above.)
